# Perinatal anxiety and compromised bond: A qualitative study of cultural scripts, structural barriers and maternal emotional negotiations in Pakistan

## Research Article

low- and middle-income countries; maternal bonding; perinatal anxiety; postpartum; qualitative research

**Corresponding author:**
Rakhshanda Liaqat;
Email: rakhshanda.phdpsy83@iiu.edu.pk

Rakhshanda Liaqat[1,2] , Kehkashan Arouj[1] and Najia Atif[2]

[1]Department of Psychology, International Islamic University Islamabad, Pakistan and [2]Human Development Research Foundation (HDRF), Pakistan

## Abstract

Maternal–infant bonding is essential for early development and long-term well-being. In low-resource settings like Pakistan, perinatal anxiety, though prevalent, remains under-recognized and can significantly disrupt bonding. While perinatal depression has garnered greater research attention, the cultural and relational dimensions linking anxiety to bonding remain under-explored. This qualitative study examined how maternal distress, sociocultural expectations and healthcare limitations influence bonding. Eighteen pregnant and postnatal women (aged 19–45 years) with clinically significant anxiety (Generalized Anxiety Disorder 7-item scale ≥ 10) were purposively recruited from public hospitals in Rawalpindi and Islamabad. In-depth interviews were conducted in Urdu and analyzed using Braun and Clarke's thematic analysis. Five major themes emerged: (1) emotional vulnerability during the perinatal period, (2) interpersonal and family dynamics, (3) maternal health and role strain, (4) cultural scripts and structural barriers and (5) participant-driven recommendations. Anxiety often delays emotional connection. Judgment, limited autonomy and lack of support worsened distress, while faith, rituals and relational coping offered resilience. This study provides novel qualitative evidence that perinatal anxiety and maternal–infant bonding are co-constructed within the relational and sociocultural ecologies of low- and middle-income countries like Pakistan. Findings challenge purely symptom-focused approaches, underscoring that effective intervention must address not only the emotional invisibility of mothers but also the relational pathways of distress, such as hypervigilance, exhaustion and performance anxiety, which are intensified by a lack of respect, autonomy and validation. A shift toward contextually grounded, relationship-centered care is urgently needed.

## Impact statement

This study provides a critical, evidence-based perspective for global mental health practitioners and policymakers working in low-resource settings. By documenting how perinatal anxiety and maternal–infant bonding are intertwined with everyday power dynamics, family pressures and systemic gaps in Pakistan, it challenges the prevailing model of care that focuses narrowly on diagnosing and treating individual symptoms. The findings offer direct guidance for creating more effective support. For clinicians and community health workers, it underscores the need to ask not just "how anxious are you?" but also "how respected and supported do you feel in your home?." For program designers, it argues that interventions will fail if they ignore the potent influence of in-laws, gendered expectations and economic stress on a mother's emotional capacity. For families and communities, the narratives redefine "help" from unsolicited advice to emotional validation and shared responsibility. Ultimately, this research advocates for a paradigm shift from viewing bonding as a private, maternal duty to recognizing it as a public health outcome that depends on a mother's social and emotional environment. Its core impact is to provide a roadmap for building maternal mental health care that is culturally grounded, relationally informed and integrated into the fabric of community and family life.





## Introduction

The development of a mother–infant bond begins during pregnancy and continues to evolve after childbirth. This process plays a fundamental role in supporting maternal adjustment and infant development. Often referred to as prenatal attachment, this relationship involves emotional connectedness, imagining the baby's future needs and even verbal communication with the unborn child (Trombetta et al., 2021). However, various emotional and psychological challenges

during the perinatal period can interfere with this bonding process (Zdolska-Wawrzkiewicz et al., 2019).

Among these challenges, perinatal anxiety is a prevalent condition characterized by persistent worry, excessive concern about fetal or infant health and somatic symptoms of tension (Folliard et al., 2020). It is conceptually distinct from perinatal depression, which is marked by persistent low mood, anhedonia and functional impairment. However, both conditions commonly co-occur and share overlapping symptoms (*e.g.*, sleep disturbance and irritability) (Rogers et al., 2020), which complicates the attribution of effects on maternal infant bonding (Nielsen-Scott et al., 2022). The chronic physiological arousal associated with anxiety can disrupt a mother's emotional availability and her capacity to attune to the infant, thereby shaping early bonding experiences (Bornstein, 2022).

This issue is particularly acute in low- and middle-income countries (LMICs), including Pakistan, where perinatal anxiety affects ~30% of mothers (Gul et al., 2019) yet remains largely unrecognized within routine care. This invisibility is not a mere clinical oversight but emerges from intersecting cultural and structural constraints (Howard and Khalifeh, 2020). Stigma surrounding mental illness, often framed as personal weakness or a threat to family honor, limits disclosure and help-seeking (Ahmad et al., 2022), and contributes to the absence of systematic screening during pregnancy and postpartum. Gendered power relations within household hierarchies, particularly the influence of in-laws and the centrality of male decision-making, further restrict maternal autonomy and intensify distress (Rowther et al., 2020). Within Pakistan's joint family system, the constant monitoring of mothers' choices and behaviors can transform caregiving into a source of anxiety rather than emotional connection (Siddiqui, 2025). These cultural pressures intersect with structural weaknesses, such as limited postpartum follow-up and the absence of maternal mental health pathways in public hospitals, to produce conditions where emotional distress remains unaddressed (Naveed et al., 2018).

Existing research from Pakistan and South Asia has effectively documented the prevalence and correlates of perinatal anxiety (Fisher et al., 2012; Gul et al., 2019; Nazir et al., 2022; Vidyasagaran et al., 2023; Waqas et al., 2023). However, this work has predominantly relied on quantitative tools that, while essential for estimating burden, do not capture the relational and emotional processes through which anxiety shapes maternal feelings and behaviors toward the infant. Despite the availability of standardized tools like the Postpartum Bonding Questionnaire (PBQ) (Brockington et al., 2001) and the Maternal Postnatal Attachment Scale (MPAS) (Condon and Corkindale, 1998), these measures have limited cultural adaptation in Pakistan, and existing literature has not explored how maternal bonding is shaped by sociocultural norms, family dynamics and co-occurring emotional distress (Khalid et al., 2019).

While intervention programs such as Happy Mother, Healthy Baby have shown promise in easing symptoms of perinatal anxiety (Surkan et al., 2024), they rarely assess changes in the emotional quality of the mother–infant relationship. When interventions focus solely on symptom reduction, they risk overlooking the subtler emotional negotiations through which mothers attempt to bond while navigating anxiety, fatigue and social pressure (Ayub et al., 2025). A qualitative approach is therefore necessary to illuminate these relational processes and to understand how anxiety interacts with cultural and structural constraints in shaping early bonding.

Guided by Bronfenbrenner's bioecological model (Bronfenbrenner and Ceci, 1994) and Mercer's theory of maternal role attainment (Mercer, 2004), this study examines maternal bonding as an evolving process of emotional negotiation shaped by sociocultural pressures, co-occurring mental health symptoms and the realities of the healthcare system in a lower-middle-income context.

## Method

### Research design

A qualitative descriptive-interpretive design was used to examine how perinatal anxiety shapes early mother–infant bonding in Pakistan. This design allowed close attention to the emotional, relational and sociocultural dimensions of bonding that cannot be fully captured through structured measures. The study forms the qualitative component of a larger mixed-methods project.

### Setting and participants

The study was conducted between January 10 and June 10, 2025, in the Obstetrics and Gynecology Departments of Holy Family Hospital, Rawalpindi, and Federal General Hospital, Islamabad, Pakistan. Participants were purposively recruited to reflect diversity across educational background, family type (joint or nuclear) and employment status. Detailed demographic and clinical characteristics of the participants are presented in Table 1.

### Recruitment and eligibility

Clinic staff directed potentially eligible women to a trained female research associate, who approached them after their clinical encounter. Screening and explanation of the study occurred in a private room. Inclusion criteria included women aged 18–45 years, currently pregnant or within 8 months postpartum, fluent in Urdu and scoring ≥10 on the Generalized Anxiety Disorder-7 scale (Spitzer et al., 2006), indicating moderate-to-severe anxiety. Exclusion criteria included active psychosis, high suicide risk, cognitive impairment or need for hospitalization. Eligible participants were approached in person at hospital outpatient clinics. Informed written consent was obtained before participation. Women were not excluded for comorbid depressive symptoms, reflecting the high overlap between perinatal anxiety and depression, and our aim to understand anxiety within naturalistic presentations. To address this analytically, we conducted a secondary coding procedure to identify depressive features. Written informed consent was obtained from all eligible participants.

### Operationalization of bonding difficulties

We did not administer a structured bonding instrument (*e.g.*, PBQ). Instead, we conceptually defined bonding difficulties through recurring narrative patterns reflecting reduced emotional connectedness, relational strain or delayed attachment. Indicators included persistent emotional "blankness," difficulty feeling affection, avoidance of physical closeness, irritability during caregiving, guilt and anxiety-driven distancing. To strengthen conceptual clarity, we developed a mapping table (Supplementary Table 1) linking these qualitative indicators to established bonding frameworks (*e.g.*, PBQ domains).

**Table 1.** Demographic and clinical characteristics of study participants (*N* = 18)

| Sr.# | Reference number | Age | Education | Family system | Number of children | Employment status | Pregnancy planned/unplanned | Age of youngest child | Marital status | Currently pregnant | Maternal stage | GAD-7 score |
|---|---|---|---|---|---|---|---|---|---|---|---|---|
| 1 | IDI–01 | 39 | Matric | Joint | 3 | Housewife | Planned | 2 years | Married | Yes | Antenatal | 10 |
| 2 | IDI–02 | 36 | Graduate | Joint | 2 | Working | Planned | 4 years | Married | Yes | Antenatal | 12 |
| 3 | IDI–03 | 31 | Intermediate | Nuclear | 2 | Housewife | Unplanned | 3 years | Married | Yes | Antenatal | 10 |
| 4 | IDI–04 | 34 | Graduate | Joint | 3 | Housewife | Planned | 6 months | Married | No | Postnatal | 10 |
| 5 | IDI–05 | 29 | Matric | Joint | 2 | Housewife | Planned | 8 months | Divorced | No | Postnatal | 16 |
| 6 | IDI–06 | 40 | Intermediate | Joint | 5 | Housewife | Unplanned | 5 years | Married | Yes | Antenatal | 10 |
| 7 | IDI–07 | 29 | Primary | Joint | 2 | Housewife | Planned | 2 years | Married | Yes | Antenatal | 10 |
| 8 | IDI–08 | 31 | Matric | Joint | 2 | Housewife | Unplanned | 3 years | Separated | Yes | Antenatal | 14 |
| 9 | IDI–09 | 41 | Matric | Nuclear | 1 | Working | Planned | 2 years | Married | Yes | Antenatal | 12 |
| 10 | IDI–10 | 45 | Middle | Joint | 7 | Housewife | Planned | 3 years | Married | Yes | Antenatal | 16 |
| 11 | IDI–11 | 22 | Graduate | Joint | 2 | Working | Postpartum | 8 months | Married | No | Postnatal | 10 |
| 12 | IDI–12 | 33 | Middle | Nuclear | 1 | Working | Planned | 2 years | Married | Yes | Antenatal | 10 |
| 13 | IDI–13 | 42 | Matric | Nuclear | 5 | Housewife | Postpartum | 3 months | Married | No | Postnatal | 12 |
| 14 | IDI–14 | 19 | Intermediate | Joint | 1 | Housewife | Planned | 2 years | Married | Yes | Antenatal | 10 |
| 15 | IDI–15 | 31 | Graduate | Joint | 2 | Housewife | Postpartum | 6 months | Married | No | Postnatal | 10 |
| 16 | IDI–16 | 41 | Primary | Joint | 4 | Housewife | Postpartum | 5 months | Married | No | Postnatal | 12 |
| 17 | IDI–17 | 21 | Middle | Joint | 3 | Housewife | Postpartum | 6 months | Married | No | Postnatal | 10 |
| 18 | IDI–18 | 34 | Matric | Joint | 3 | Housewife | Postpartum | 4 months | Married | No | Postnatal | 10 |

*Note*: GAD-7, Generalized Anxiety Disorder 7-item scale; antenatal, during pregnancy; postnatal, after childbirth.

### Data collection

Data were collected through semi-structured in-depth interviews (IDIs) with 18 women, conducted in Urdu by two trained female research associates. The interview guide (see Supplementary Material), pilot-tested and refined for cultural relevance, covered: emotional changes, anxiety triggers, feelings toward the infant, caregiving experiences, sociocultural pressures and support systems. Interviews lasted 45–60 min, were audio-recorded with consent and followed a strict distress protocol with an on-site psychologist available.

### Data management and translation

Interviews were transcribed verbatim in Urdu and analyzed in the source language to preserve meaning. The excerpts selected for presentation underwent a structured translation process: bilingual team members translated them into English; a random 20% sample was independently back-translated to Urdu to check for discrepancies, which were resolved in team consensus meetings. Culturally specific terms (*e.g.*, *chilla*, the 40-day postpartum rest period) were retained with explanatory notes.

### Data analysis

Data collection and analysis were conducted concurrently to ensure an iterative and reflexive approach. Thematic analysis followed Braun and Clarke's framework (Braun and Clarke, 2006). Two researchers independently conducted initial open coding in both Urdu and English. Codes were documented in a shared Excel codebook and refined through repeated comparison within and across transcripts, gradually developing into broader categories and themes.

To ensure analytic rigor, 25% of transcripts were double-coded, and discrepancies were resolved through discussion or adjudication by a senior qualitative researcher. Analytic memos, team debriefings and systematic within-case and cross-case comparisons were used throughout the process. A secondary, targeted coding cycle was applied to identify depressive features such as emotional blunting or pervasive guilt to account for the natural overlap between anxiety and depression in perinatal populations. All analytic decisions were documented to maintain auditability and transparency.

### Reflexivity, positionality and trustworthiness

The research team comprised Pakistani female researchers trained in maternal mental health. Reflexive discussions were held throughout data collection and analysis to address assumptions about sociocultural norms, family roles and emotional expression. Trustworthiness was enhanced through triangulation (interviews and field notes), team-based coding, detailed documentation of analytic decisions and the inclusion of deviant or contradictory cases.

### Sample size and thematic saturation

Eighteen interviews were conducted. Code saturation was assessed throughout the analysis. The coding team documented a sharp decline in new codes by interview 15, and no new higher-order themes emerged by interview 18, indicating thematic saturation.

### Ethical considerations

Ethical approval was obtained from the Institutional Review Board of Islamic International University, Islamabad (Ref: IIU/2025-DGS

8569). Written informed consent was obtained from all participants. Women were informed about the voluntary nature of participation and their right to withdraw at any time without consequence. All data were de-identified using pseudonyms. Audio and transcript files were securely stored on encrypted servers. Counseling support was available for participants who experienced distress during or after the interview process.

## Results

We interviewed 18 women. Our analysis generated five overarching themes, each with multiple subthemes, illustrating the complex and multidimensional ways perinatal anxiety shapes maternal–infant bonding in Pakistan. Participants described experiences marked by emotional strain, restricted autonomy and intense caregiving expectations, but they also shared moments of resilience and intentional connection. These narratives show that bonding is not an isolated psychological process; it is shaped by the combined influence of anxiety symptoms, family dynamics, cultural norms and structural constraints. A full thematic overview is presented in Table 2.

### Theme 1: Emotional states and psychological vulnerability

#### Subtheme 1.1: Anxiety and hypervigilance

Many participants experienced anxiety as pervasive hypervigilance, which directly consumed the mental space needed for emotional connection. This often manifested as compulsive checking and intrusive fears about infant health. For instance, one mother described: "I changed his clothes five times a day… this affected both his and my health" (IDI-05). Another participant, adjusting to new motherhood, described constant worry: "Even at night, I would wake up frequently to check if my baby was breathing properly. I never left my baby alone, not even for a minute. I had this fear that something bad would happen" (IDI-02). Fear extended beyond the home into routine antenatal visits: "Whenever I went for my antenatal check-ups, I felt scared, wondering what the doctor would say about whether the baby was healthy or not. I was constantly anxious about my baby's growth and wellbeing. Every little thing would make me worry" (IDI-04). This hypervigilance, a core feature of anxiety, monopolized attentional resources that might otherwise have been directed toward quiet, attuned bonding.

#### Subtheme 1.2: Emotional dissonance, unpreparedness and depressive feelings

For many women, anxiety blended with sadness, guilt and emotional numbness, creating an early sense of disconnection from the infant. This emotional dissonance was often rooted in limited reproductive autonomy, particularly when pregnancies were unplanned or externally imposed. Mothers described feeling guilty for not experiencing the expected joy: "I felt guilty for not feeling the happiness I thought I should after giving birth" (IDI-12). Others spoke of numbness and self-doubt: "I didn't cry tears of joy like in dramas… I just felt blank. I thought something was wrong with me" (IDI-06). For some, the lack of decision-making power intensified this disconnect: "This wasn't my decision… they wanted another child. I wasn't ready" (IDI-08). In more pronounced cases, women described profound detachment: "I couldn't even look at my baby without crying. I didn't feel like a mother" (IDI-07), reflecting an overlap of anxiety with depressive feelings that created a substantial barrier to early bonding.

#### Subtheme 1.3: Trauma exposure, abuse and emotional withdrawal

Several participants described traumatic experiences that damaged and destabilized the emotional foundation needed for bonding. Intimate partner violence, coercive control or major family instability created fear that made connecting with the baby feel impossible. One woman explained, "I was divorced in my seventh month. I didn't feel any connection to the baby" (IDI-05). Another who experienced physical abuse said, "My husband used to beat me, even during pregnancy. How could I feel happy about this child when I was constantly afraid?" (IDI-08).

Some mothers interpreted their emotional detachment through culturally embedded explanations, such as beliefs about spiritual harm: "Because of black magic, I don't feel any love for my children… I don't feel any attachment with the baby in my womb either" (IDI-10). Others linked emotional distance to chronic disrespect at home: "When the situation at home is not good and the wife is treated like a doormat, a normal relationship with the child cannot develop" (IDI-13). Participants also noted how violence harms children themselves: "In our neighborhood, a woman faced domestic violence. Her children were deeply affected they would faint, live in fear, and act out" (IDI-09).

#### Subtheme 1.4: Resilience and emotional regulation in pregnancy

Despite their emotional challenges, many women showed remarkable resilience, often drawing strength from faith, routine and social support. A 39-year-old mother of three said:

"Staying relaxed and doing prayers made delivery smooth" (IDI-01). A participant who coped through spiritual practice

**Table 2.** Emergent themes and subthemes from thematic analysis

| Main theme | Subthemes |
|---|---|
| Theme 1: Emotional States and Psychological Vulnerability During the Perinatal Period | 1.1 Anxiety and Hypervigilance 1.2 Emotional Dissonance, Unpreparedness and Depressive Feelings 1.3 Trauma Exposure, Abuse and Emotional Withdrawal 1.4 Resilience and Emotional Regulation in Pregnancy |
| Theme 2: Interpersonal and Familial Dynamics | 2.1 Marital Relationships: Support, Distance and Abuse 2.2 Joint Family System- Support, Interference, and Emotional Strain 2.3: Gender Preference and Structural Devaluation of Daughters 2.4 Maternal Autonomy *Versus* Social Surveillance |
| Theme 3: Maternal Health and Role Exhaustion | 3.1 No Time for Healing and Connecting With the Baby 3.2 Postnatal Emotional Exhaustion and Burnout 3.3 Work Life Conflict and Guilt in Employed Mothers |
| Theme 4: Cultural Scripts and Structural Barriers | 4.1 Idealized Motherhood and Cultural Stigmatization 4.2 Financial Strain and Healthcare Access 4.3 Traditional Postpartum Practices and Cultural Constraints 4.4 Bonding as a Gradual and Non-Linear Process 4.5 Parenting Medically Vulnerable Children |
| Theme 5: Participant-Driven Recommendations | 5.1 Enhanced Emotional Support Systems 5.2 Improved Healthcare Access and Education 5.3 Cultural Shifts in Family Dynamics |

shared: "I used to walk every evening and listen to Quran. It gave me peace, and I felt more present with my baby" (IDI-13). Another woman, isolated in her daily routine, found comfort in connection: "My cousin used to call every day. Just talking to her helped me breathe. I felt less alone" (IDI-11). These efforts were crucial for mitigating the disruptive impact of anxiety on bonding.

### Theme 2: Dynamics of interpersonal and broader familial relationships

#### Subtheme 2.1: Marital relationships support, distance and abuse

While trauma and abuse are addressed in Theme 1 as sources of psychological vulnerability, this subtheme focuses specifically on the ongoing relational dynamics within marriage and how everyday patterns of support, neglect or distance shaped mothers' emotional capacity during the perinatal period. Spousal support emerged as a critical buffer for some, while its absence was a potent stressor for others. The spectrum ranged from stabilizing emotional partnership to profound neglect, and in severe cases, to intimate partner violence, which constituted a distinct context of terror that fundamentally severed psychological safety. A 33-year-old graduate mother explained how her husband's emotional presence helped her navigate this vulnerable time: "My husband's emotional and physical support helped me stay mentally healthy" (IDI-03). In contrast, others shared experiences of emotional distance and neglect. A 31-year-old participant recalled: "He became so distant… I started questioning my worth. I felt alone in this journey. It affected my ability to connect with my baby" (IDI-02). Some women, like a 29-year-old mother of two, described how a lack of empathy eroded communication: "He never asked how I was. I stopped sharing" (IDI-18).

#### Subtheme 2.2: Joint family system – support, interference and emotional strain

Experiences within the joint family system ranged from protective and nurturing to intrusive and destabilizing. Some women described receiving essential postpartum care: A participant recovering from C-section said her in-laws' help was lifesaving: "Without my mother-in-law's help after my surgery, I wouldn't have survived. She took care of the baby when I couldn't even stand" (IDI-14). At the same time, many participants described constant criticism and unsolicited advice that undermined confidence: "My in-laws criticized everything from how I held the baby to how much I breastfed. I started hiding things, stopped asking for help" (IDI-08). For a few women, the presence of family members felt more performative than supportive: "They were always around, but never truly present. I felt more like a servant than a new mother" (IDI-06). Family involvement sometimes spilled into marital tension: "In-laws are always trying to create conflict between husband and wife. In such an environment, even if a mother is attached to her child, it is not enough. A child needs both parents" (IDI-04).

#### Subtheme 2.3: Gender preference and structural devaluation of daughters

Gender preference was a recurring theme, reflecting broader social norms rather than isolated personal attitudes. Many women linked their emotional distress to how pregnancies and postpartum care were treated differently depending on the infant's gender.

A woman shared the pain of feeling devalued as a mother to daughters: "People would say, 'Insha'Allah it will be a boy this time'… No one thought about how it made me feel. They acted like my daughters are a burden" (IDI-10). Another reflected on clear differences in care and respect: "If a son is born, the husband and in-laws take better care of the mother, they give her more rest and importance" (IDI-08). Unequal postpartum treatment based on the baby's sex was commonly reported. A mother recalled: "After my daughter's birth, I was expected to return to work and chores quickly. But when my sister-in-law had a boy, they let her rest for weeks" (IDI-12). These narratives highlight how structural gender bias places additional emotional demands on mothers of daughters, shaping their sense of worth and influencing the emotional space available for bonding.

#### Subtheme 2.4: Maternal autonomy versus social surveillance

Many women described how their roles as mothers were closely monitored and routinely judged. One participant recalled: "Everyone has a problem with the way I mother. I cannot laugh, cry, or even hold my baby my way. I'm a mother, but I'm not free" (IDI-17). "I would get phone calls during hospital visits asking when I'd return. They didn't care I was waiting for my turn for a check-up" (IDI-04). These accounts demonstrate how intrusive oversight within the joint family system can undermine mothers' confidence, contributing to stress and emotional exhaustion during the critical bonding period.

### Theme 3: Maternal health and role exhaustion

#### Subtheme 3.1: No time for healing and connecting with the baby

The postpartum period, often portrayed as a sacred time for bonding and recovery, was for many participants marked by physical strain and relentless responsibilities. A mother of two living in a joint family system shared, "I couldn't sit properly for weeks after the birth. But I had to keep getting up to feed and clean. It wasn't bonding it was surviving" (IDI-05). Another participant recalled, "They said, [In-laws] 'You're young, you'll heal fast.' But I was in pain, and they expected me to act normal. I couldn't even hold my baby properly without wincing" (IDI-02).

#### Subtheme 3.2: Postnatal emotional exhaustion and burnout

Many women, particularly those without sustained support, described profound emotional depletion, sensory overload and persistent exhaustion. One working mother shared, "I would come home after work, exhausted, and still be expected to do everything. No one asked if I had eaten, if I had rested" (IDI-09). For some, this relentless fatigue escalated into irritability and self-doubt. A mother of three living in a joint family system reflected, "I became very irritated… even felt angry toward my baby. Then I felt guilty for being that way" (IDI-17). These accounts demonstrate how continuous demands, sensory fatigue and emotional invisibility produced burnout that eroded mothers' emotional availability and strained early bonding.

#### Subtheme 3.3: Work–life conflict and guilt in employed mothers

Mothers who returned to work described the tension between financial necessity, caregiving expectations and emotional attachment, often navigating cultural judgments alongside personal guilt. Their professional responsibilities often clashed with societal expectations and personal desires to remain close to their infants. A 36-year-old working mother shared, "No one asks why she has to work… they just blame her. They say the baby is crying because the mother is selfish" (IDI-02). Another woman, working part-time, remembered, "I left my daughter with my sister. When I came home, they said, 'she (daughter) was happy staying with us, she didn't even look for you.' That broke me" (IDI-12). A third

participant, who returned to work only 2 months postpartum, admitted, "I cried in the bathroom at work. I felt like I was abandoning her. But if I didn't earn, how would we survive?" (IDI-04).

### Theme 4: Cultural scripts and structural barriers

#### Subtheme 4.1: Idealized motherhood and cultural stigmatization

Many participants described intense pressure to conform to idealized expectations of motherhood, where emotional struggle was equated with weakness. A 31-year-old postnatal mother recalled, "When I cried in front of my mother-in-law, trying to explain that doing home chores with a baby is very challenging, she said I was being weak and ungrateful" (IDI-18). Another participant explained, "In my husband's family, there's a strong belief that suffering makes you a better mother. So if a woman speaks up, it means she's not strong enough" (IDI-02). As one mother stated, "Mothers are expected to sacrifice everything, and if they complain, they are judged" (IDI-08). These cultural expectations often silenced women and discouraged open discussion of emotional distress, constraining early bonding and shaping how mothers experienced their roles.

#### Subtheme 4.2: Financial strain and inequitable healthcare access

Participants consistently described how financial barriers and systemic limitations shaped their ability to access even the most basic maternal healthcare. A mother of three emphasized how basic medical support felt out of reach: "Checkups are a luxury… how you can even think about emotions?" (IDI-04). Another woman described the tension that financial hardship created in her household: "My husband used to fight with me about transport money. Even with the hospital allowance, it was not enough. So I missed appointments" (IDI-17). Several participants noted that even when they reached healthcare facilities, the experience was disheartening. A 29-year-old antenatal mother shared: "We wait for hours, and then they rush us. No one asks how we feel emotionally." (IDI-10). Others highlighted how perceptions of neglect within the system deepened their distress: "They only care if something is physically wrong. If you're crying or anxious, they just say: 'It's normal.'" (IDI-07).

#### Subtheme 4.3: Traditional postpartum practices and cultural constraints

Most participants described traditional postpartum confinement practices, such as the 40-day chilla, dietary restrictions and strict limitations on movement, as emotionally stifling rather than restorative. These rules, typically enforced by elder women, often curtailed autonomy and intensified anxiety. One mother shared, "They restricted me for 40 days. I wasn't allowed to go out or meet my friend. I felt bored and depressed… It felt like prison" (IDI-03). Another recalled, "My mother-in-law said take small sips or the baby would get colic, but who cared if I was hungry? I wanted food, not rituals" (IDI-09). Stigma around childbirth mode also surfaced: "They said I didn't experience 'real' delivery because I had a C-section. I felt ashamed of something I couldn't control" (IDI-02). These accounts illustrate how rigid postpartum expectations can undermine emotional comfort, restrict self-care and complicate early bonding.

#### Subtheme 4.4: Bonding as a gradual and nonlinear process

Many participants described bonding not as an immediate emotion, but as a gradual, uneven process shaped by physical recovery, emotional strain and external pressures. A mother of three reflected, "I bonded immediately with my first… but after the cesarean with my second, it felt like something was missing. I just felt empty" (IDI-13). Another shared, "It took weeks to feel like a mother. I fed him, changed him… but emotionally I was far away. No one tells you that's normal" (IDI-12). For some, their own childhood histories shaped this trajectory. One woman explained, "When I was a child, I used to wish my mother would hug me… I did the complete opposite with my baby" (IDI-04). Together, these accounts emphasize that bonding unfolds over time and is deeply influenced by both past and present emotional realities, especially when mothers are recovering from childbirth or navigating anxiety.

#### Subtheme 4.5: Parenting medically vulnerable or neurodivergent children

For mothers of infants with medical complications or developmental conditions, the early postpartum experience was marked by added stress, stigma and emotional fatigue. One participant recalled, "They said my child's autism was punishment of our sins… no one truly understands" (IDI-16). Another woman described the judgment she faced, "When my child was diagnosed with special needs, people kept whispering behind my back, saying that it's my fault. It was hard to handle the judgment while I was just trying to adjust" (IDI-04).

### Theme 5: Participant-driven recommendations

#### Subtheme 5.1: Enhanced emotional support systems

Many participants emphasized that what they needed most, more than advice or suggestions, was kindness, emotional safety and a sense of being seen. A participant from a joint family system expressed: "Mothers need support, not judgment… A kind word can change her experience" (IDI-09). This sentiment was echoed in the desire for a more empathetic societal lens toward motherhood. "We need to teach society that motherhood is not a competition. It's not about who suffered more, but who got supported more" (IDI-16). These recommendations highlight a central theme across interviews: emotional safety and compassionate communication are critical enablers of maternal well-being and early bonding.

#### Subtheme 5.2: Improved healthcare access and education

Participants called for stronger, more accessible healthcare systems and greater awareness of maternal mental health, noting how fragmented care contributed to emotional strain.

Several women called for stronger healthcare systems and education around maternal mental health. A young mother of one emphasized: "Better prenatal care and awareness could help mothers feel more prepared for bonding" (IDI-04). Another participant, a 45-year-old mother of seven daughters, explained the cumulative toll of health neglect: "Postnatal care is just as important as prenatal. We feel weak, overwhelmed… without medical guidance, we suffer silently" (IDI-10). Participants repeatedly highlighted the need for health workers to go beyond physical check-ups, creating space for emotional check-ins and referrals when needed.

#### Subtheme 5.3: Cultural shifts in family dynamics

Participants highlighted how cultural norms around childbirth often overlook mothers' emotional needs. As one woman explained, "People make visits after birth out of curiosity to see how the baby looks like, but no one really cares about how the mother is doing." (IDI-02). Many women connected these

dynamics to broader generational patterns and expressed a desire for change. One mother reflected, "As we saw our mothers suffering in silence bearing the pressures of society, we have to make sure to end this cycle by giving your children love and security, not burdens" (IDI-18). Together, these accounts show mothers' awareness of how entrenched family norms shape emotional climates and their belief that cultural shifts are essential for healthier bonding and long-term child well-being.

## Discussion

This qualitative study reframes perinatal anxiety from an individual psychological symptom to a relational process embedded within the sociocultural ecology of Pakistan. By examining how anxiety shapes early maternal–infant relational experiences, our findings reveal that bonding difficulties are not maternal deficits but outcomes of relational strain within gendered, resource-poor settings.

Our findings extend perinatal mental health literature in LMICs by detailing anxiety's relational manifestations. In contrast to approaches that treat anxiety as an internal psychological state, participants experienced it through patterns that disrupted the caregiving environment: hypervigilance consumed attentional resources, emotional withdrawal created relational distance and performance anxiety under family surveillance stifled spontaneous interaction. A consistent pattern was the co-occurrence of anxious and depressive features, emotional numbness, guilt and irritability. This symptom blending reflects how structural stressors blur traditional diagnostic categories and produce a spectrum of distress that is relationally experienced. The clinical distinction between anxiety and depression may be less meaningful in this context than understanding their combined impact on maternal availability and bonding, a pattern noted in prior perinatal mental health research (Falah-Hassani et al., 2017; Gul et al., 2019; Kalin, 2020).

Methodologically, our qualitative approach was essential for capturing these relational nuances. While standardized tools such as the PBQ (Brockington et al., 2001) and MPAS (Condon and Corkindale, 1998) are valuable for screening, their limited cultural adaptation in Pakistan risks overlooking how bonding is shaped by family dynamics, gendered expectations and co-occurring emotional distress (Khalid et al., 2019). By focusing on mothers' narratives, we moved beyond quantifying bonding to understanding its relational and contextual mediators. This revealed that anxiety affects bonding not merely through internal worry, but through interpersonal pathways such as attentional drain, emotional exhaustion and performance pressure.

The data demonstrate how socio-structural pressures translate into relational anxiety. Financial strain, gendered expectations and fragmented healthcare do not merely correlate with anxiety; they produce it by constraining mothers' autonomy, rest and emotional safety. Participants internalized these macro-level conditions through relational negotiations: fear of judgment from in-laws, guilt over unmet caregiving ideals and exhaustion from uncompensated labor. This embodied stress then restricted emotional presence, creating a feedback loop where structural inequities become lived as relational withdrawal. Our findings thus extend LMIC literature on social determinants of maternal mental health (Rowther et al., 2020; Insan et al., 2022) by mapping the mechanism through which structure becomes emotion and how that emotion, in turn, shapes early bonding (Gholampour et al., 2020).

Cultural practices, such as *chilla* (the 40-day postpartum rest period), further illustrated this tension between support and control. While some women found the structure protective, others experienced it as coercive, especially when rituals were rigidly enforced without regard for maternal preference or well-being. This negotiation of tradition highlights women's constrained agency: even culturally sanctioned support could heighten anxiety when it limited autonomy, prioritized ritual over need or became a means of familial surveillance. Such nuanced findings caution against dichotomizing cultural practices as either "protective" or "restrictive," and instead call for approaches that respect maternal voice within traditional frameworks, consistent with ambivalent findings on postpartum rituals in South Asia (Qamar, 2017; Ahmed et al., 2020; Vernekar, 2021).

Theoretically, our findings both align with and complicate established models. They support Bronfenbrenner's bioecological framework (Bronfenbrenner and Ceci, 1994) by illustrating how anxiety arises from interactions across micro-system (marital tension), meso-system (family-healthcare interactions) and macro-system (gender norms and poverty) levels. Notably, the constant monitoring by family members represents a form of relational pressure not fully captured in traditional ecological schemas. Similarly, Mercer's theory of maternal role attainment (Mercer, 2004) helps explain the nonlinear, often delayed bonding described by participants, particularly when anxiety and lack of support disrupted the transition to maternal identity. However, Mercer's focus on individual adaptation requires expansion to account for how structural and interpersonal hostility can actively inhibit "becoming a mother," especially in contexts where motherhood is idealized yet unsupported, echoing broader work on maternal identity formation under constraint (Ali et al., 2023; Kaundal and Itoo, 2024; Hussain and Usman, 2025; Sharif and Sabir, 2025).

Taken together, this study offers three distinct contributions to global mental health research and practice. First, it conceptualizes perinatal anxiety as a relational phenomenon, demonstrating how it disrupts bonding through specific interpersonal patterns, hypervigilance, emotional withdrawal and performance anxiety, rather than solely through intrapsychic distress. Second, it maps the pathways through which sociocultural structures mediate bonding, showing how financial precarity, gendered surveillance and fragmented healthcare reduce maternal autonomy, rest and emotional safety, thereby straining the mother–infant relationship. Third, it identifies respect and equitable care as critical, under-theorized buffers against anxiety's relational impacts. In settings where mothers are often undervalued, respect expressed through emotional validation, shared responsibility and supportive partnership emerged as a foundational resource for mitigating distress and enabling connection, complementing broader frameworks for contextual and relationally grounded care in LMICs (Maselko et al., 2015; Sikander et al., 2019; Surkan et al., 2024).

These insights underscore the need for maternal mental health approaches that address not only symptoms but the relational and structural conditions under which caregiving occurs. Effective support in LMIC settings like Pakistan will require moving beyond clinic-centered, individually focused models toward ecologically informed, relationally attuned care that engages families, communities and health systems as partners in maternal well-being.

## Limitations

This study has several limitations. First, the sample was drawn from a single urban public hospital, which may limit transferability to women giving birth at home, in private facilities or in rural areas where patterns of support and access differ. Second, all interviews

were cross-sectional, restricting insight into how anxiety and bonding evolve over time; longitudinal follow-up may have clarified whether the emotional distance some mothers described persisted or resolved. Third, the interviewer participant power dynamics may have influenced disclosure, as cultural norms of deference and emotional restraint could shape how openly women communicated distress, despite reflexive practices to mitigate these effects. Fourth, translation from Urdu to English, although conducted through a structured, bilingual process, may not fully capture all cultural and emotional nuance. Finally, because depression was not formally screened, some women may have had overlapping depressive features, and bonding was explored qualitatively rather than through diagnostic tools; thus, findings reflect subjective relational experiences rather than clinical impairment.

### Implications for practice and policy

The findings point to several practical implications for maternal mental health care in Pakistan and similar lower-resource settings.

#### Clinical practice

Routine antenatal and postnatal screening should include brief anxiety tools and relational questions about the mother–infant connection. Task-sharing models that train nonspecialist providers (*e.g.*, lady health workers) to deliver culturally adapted psychological support can expand access, as demonstrated in Pakistan. Clinical environments must also prioritize privacy, supportive interaction and clear referral pathways for intimate partner violence or severe distress, enabled by stronger interprofessional collaboration.

#### Intervention design

Existing programs (*e.g.*, *Happy Mother, Healthy Baby*) should integrate modules on mother–infant relational experiences, addressing anxiety-driven hypervigilance, emotional overload and cultural ideals of motherhood. Family-inclusive psychoeducation, particularly for husbands and mothers-in-law, can help reduce criticism, surveillance and gendered pressure.

#### Policy

Structural investment is needed. Policies supporting maternity leave, stigma reduction and social protection can alleviate chronic stressors that undermine bonding. Training community health workers to offer basic mental health support and peer groups can provide mothers with relational validation beyond restrictive household settings.

### Conclusion

This study provides qualitative insight into how perinatal anxiety shapes maternal–infant relational experiences within the cultural, familial and structural context of Pakistan. The findings demonstrate that bonding difficulties are not merely individual psychological phenomena but emerge from a broader ecology in which anxiety, gender norms, household dynamics and limited health system support interact.

By centering women's narratives, the study highlights that anxiety affects bonding through relational pathways such as hypervigilance, emotional exhaustion and performance pressure, all intensified when mothers lack respect, autonomy or emotional validation. These insights underscore the need for maternal mental health approaches that address not only symptoms but the relational and sociocultural conditions under which caregiving occurs.

Supporting maternal–infant bonding in Pakistan requires shifting from a narrow, clinic-centered model to one that recognizes mothers' emotional needs as embedded within families, communities and health systems. Future research should adopt longitudinal and mixed-methods designs to map how bonds develop over time and how interventions can strengthen relational security in settings marked by gendered and structural inequities.

**Open peer review.** To view the open peer review materials for this article, please visit http://doi.org/10.1017/gmh.2026.10152.

**Supplementary material.** The supplementary material for this article can be found at http://doi.org/10.1017/gmh.2026.10152.

**Data availability statement.** Due to the sensitive nature of the interviews and confidentiality agreements with participants, the qualitative transcripts are not publicly available. However, de-identified excerpts or further information may be made available upon reasonable request to the corresponding author.

**Acknowledgments.** The authors would like to thank the Human Development Research Foundation for providing field and logistical support for this study. We are especially grateful to the two research associates who helped conduct the interviews with sensitivity and dedication.

**Author contribution.** Rakhshanda Liaqat conceived the study, led data collection and analysis, and drafted the manuscript. Kehkashan Arouj supervised the study and assisted with manuscript review. Najia Atif provided overall mentorship and oversight for fieldwork and contributed to data analysis, including refinement of themes and subthemes.

**Financial support.** This research received no specific grant from any funding agency in the public, commercial or not-for-profit sectors.

**Competing interests.** The authors declare none.

**Ethics statement.** This research adhered to strict ethical principles and guidelines throughout its conduct. Informed consent was obtained from all participants, ensuring they fully understood the study's objectives, procedures and their right to withdraw at any time without any consequences. To protect participants' privacy, all transcripts were anonymized, and personally identifiable information was removed. Ethical approval was granted by the Institutional Review Board of the International Islamic University Islamabad (Ref: IIU/2025-DGS 8569). Given the sensitive nature of the topic, particular care was taken to ensure participant comfort and emotional safety during interviews. This study reflects a strong commitment to upholding ethical standards, safeguarding the rights and well-being of participants and contributing valuable insights into prenatal and postnatal mental health in resource-constrained settings.

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
