## [Reviewer Report]

Anxiety is indeed highly prevalent during the perinatal period, so this this paper´s goal addresses a very imortiant question regarding interference with mother infant bonding due to anxiety. We have some comments:

1. Material and methods:

2. How was the “compromised bonding” measured?

There are scales for bonding, including one developed by Brockington (Brockington IF, Fraser C, Wilson D. The Postpartum Bonding Questionnaire: a validation. Arch Womens Ment Health. 2006 Sep;9(5):233-42. doi: 10.1007/s00737-006-0132-1. Epub 2006 May 4. PMID: 16673041.) (The authors quote prof Brockington )

3. About the selection criteria: the women should’ve been screened for depression. The narratives presented, in some cases, are suggestive of depression. For instance, subtheme 3. 2 postnatal exhaustion and burnout. Women describe fatigue, lack of interest in self care, being tired and then irritable, with guilt feelings. The same goes for the feelings of emptiness described in subtheme 4.4

Anxiety and depression do go together sometimes, but then, in order to understand the (relative) contribution of anxiety to compromised bonding, women with depressive symptoms need to be grouped apart. Otherwise, results will be difficult to analyse.

4. Material and methods need to be sorted out more clearly: the number of women interviewed should be mentioned in the results (line 133)

5. Likewise Table 2 belongs in the Results section

6. Then the authors can start the Results with a statement such as, “ We interviewed 18 women”

7. As stated before, several narrative excerpts presented seem to correspond to depression and anxiety rather than anxiety alone.

8. The discussion assumes that these women were not capable of ideal bonding with their babies, but there is no demonstration of it because there is no measure or evaluation of bonding

9. The cultural issues described are a painful reality in Pakistan and other countries, where they have unfavorable consequences on the mental health of women and there is a dire need for change. Mental health researchers can actively contribute with solid evidence.

10. I don’t think the conclusion is supported by the data

---

## [Reviewer Report]

Introduction:

Second paragraph: Readers may benefit from having definitions of perinatal anxiety and postnatal depression and some additional explanation of the differences (and perhaps, similarities) between both conditions. This reviewer is wondering about the possible overlap of symptoms and the possibility of a patient having both conditions concurrently, and in such situations, the impact of having one condition on the other.

Third paragraph: The authors articulately describe the factors which lead to perinatal anxiety in Pakistan is underdiagnosed. This reviewer wonders if there is room to discuss cultural norms around mental health. How are mental health and mental illness accepted in the everyday experience and discourse? How much stigma is attached to mental illness? Lines 90-91 hint at the level of stigma. What are the reasons why routine screening for maternal mental health not standard practice? And does this refer to mental health screening during pregnancy or postpartum, or both?

This reviewer wonders also if there is room to include the dynamics of power, as the experiences of mothers described in this paragraph speak to cultural expectations and norms which are enabled by power and privilege. These cultural narratives are confining and disempowering for mothers, and yet, they find ways to reclaim their agency. Their acts of subversive agency should be appreciated and contextualized.

Fourth paragraph: are there tools that have been developed in LMICs? And have the BPQ or the MPAS been translated into Urdu?

This reviewer wonders if the authors would consider including a statement of positionality.

Methods:

Exclusion criteria: while active psychosis and high suicide risk are included in the exclusion criteria, was depression screened for in participants and would they have been excluded?

Could the interview questions be included?

Results:

Subtheme 1.2:

Please see comments regarding the second paragraph of the Introduction. This reviewer feels that there is likely overlap between perinatal anxiety and postnatal depression and this should be discussed.

Subtheme 1.3/Subtheme 2.1:

This reviewer works with many patients who are survivors of Intimate Partner Violence (IPV), and wonders if this is a distinct experience from other harms, including marital disharmony and spiritual harm. Children who witness IPV in the home sustain significant harm even though they may not be recipients of direct abuse.

Subtheme 2.3:

Cultural bias favouring male children extends to the care extended to the mothers of male children, and this reviewer wonders if this dynamic reflects the wider cultural “value” assigned to being male and the devaluation of being female, which extends to mothers.

Discussion:

One experience that the authors have not included and might consider including in their Discussion is that of respect. The mothers who felt supported in the postpartum period were respected as partners, as mothers and as people who were capable. They were respected and valued, beyond their maternal roles. The mothers who felt unsupported were disrespected and their rights as persons, partners and mothers were disregarded.

Overall, this manuscript was enjoyable and educational for this reviewer to read. Kudos to the authors for sharing the experiences of mothers in Pakistan and for advocating for “culturally grounded, relationship-focused approaches to maternal mental health in low- and middle-income countries.”

---

## [Reviewer Report]

Perinatal Anxiety and Compromised Bond: A Qualitative Study of Cultural Scripts, Structural Barriers, and Maternal Emotional Negotiations in Pakistan

1. Abstract

Ensure the conclusion highlights the study’s novelty within LMIC contexts.

2. Introduction

• Strengthen the narrative connection between perinatal anxiety and bonding difficulties; some sections read as parallel rather than integrated.

• Reduce redundancy around cultural influences (family scrutiny, postpartum rituals) and organize them more cohesively.

• Expand on how existing Pakistani or South Asian literature has not addressed the relational/emotional dimensions of bonding.

• Provide a sharper rationale for why a qualitative design is required for this topic.

• More clearly articulate the research gap and how this study fills it.

3. Methods

• Clarify who approached potential participants and at which point in clinical encounters screening occurred.

• Provide justification for sample size in relation to thematic saturation.

• Ensure fragmented sentences are corrected for clarity.

• Give more detail on the translation process (Urdu → English) and how linguistic meaning was preserved.

• Specify whether any software was used

• Strengthen explanation of how the coding framework evolved through iteration.

4. Results / Findings

• Consider condensing repetitive quotes and combining overlapping subthemes for clarity.

• More explicitly quantify representation (e.g., “most participants,” “several participants”) without implying statistical generalization.

• Improve consistency in how participant identifiers (e.g., Mother-03) are formatted.

• Strengthen analytic commentary that accompanies quotes to enhance interpretive depth.

5. Discussion

• Improve overall structure: clearly separate interpretation, theoretical implications, cross-study comparisons, and contributions.

• Clarify how the study extends findings from previous perinatal mental health research in LMICs.

• Adopt a more analytical tone—some sections read as overly emotive rather than interpretive.

• Ensure the narrative clearly demonstrates how qualitative insights contribute uniquely to global mental health.

6. Limitations

• Broaden discussion to include how hospital-based sampling may exclude home births and rural populations.

• Note potential interviewer–participant power dynamics, especially given cultural sensitivities.

• Acknowledge limitations related to cross-sectional timing and lack of follow-up.

• Reflect briefly on potential bias introduced through translation and transcription.

7. Implications for Practice and Policy

• Provide more specific, actionable recommendations for health systems, such as clinical pathways, screening protocols, or family-inclusive practices.

• Clarify how findings can inform design or adaptation of existing interventions (e.g., Happy Mother, Healthy Baby).

• Expand discussion on how culturally grounded approaches could be operationalized in overstretched public facilities.

8. Conclusion

• Strengthen the final claim about the study’s contribution to global mental health literature.

• Reiterate the conceptual link between perinatal anxiety, relational dynamics, and bonding difficulties succinctly.

• Ensure language is precise and avoids overgeneralization beyond the study sample.

---

## [Reviewer Report]

Thank you for the opportunity to review this manuscript. This qualitative study addresses an important aspect of maternal bonding by mothers in Pakistan, i.e., maternal distress intersects with social and cultural expectations and norms, and limitations of healthcare services and delivery. Kudos to the authors for sharing the experiences of mothers in Pakistan and for advocating for “culturally grounded, relationship-focused approaches to maternal mental health in low- and middle-income countries.”

Overall, this manuscript was enjoyable and educational for this reviewer to read. The authors have completed a major revision of their original draft with improvements in clarity and additions of explanatory details.

Results

Subtheme 1.2: Emotional Dissonance, Unpreparedness and Depressive Affect

This reviewer wonders about the inclusion of “depressive affect” as it describes the outward, external observable emotional expression, whereas the other descriptors relate to the women’s feelings and internal experiences. It is also a term used by medical/health professionals to describe the physical appearance of a patient with depression.

Subtheme 1.3: Trauma Exposure, Abuse, and Emotional Withdrawal

This reviewer would suggest that trauma does not break the emotional foundation needed for bonding, but rather damages the emotional foundation and disrupts the bonding process.

Lines 366-367 which speak about spiritual harm seem out of place, unless the authors are suggesting that these women interpret IPV as a form of spiritual harm.

Subtheme 2.1: Marital Relationships, Support, Distance and Abuse

There is overlap between subtheme 1.3 and 2.1: how did the authors decide that these overlapping subthemes were distinct?

Subtheme 2.3: Gender preference

The authors should not use “gender” to replace “sex”. For example, “[u]nequal postpartum treatment based on the baby’s gender was commonly reported” should be “baby’s sex”.

---

## [Reviewer Report]

Thank you for your careful and thoughtful revisions. The authors have adequately addressed all the comments and recommendations from the previous review. The revisions have improved the clarity, rigour, and overall quality of the manuscript. I have no further substantive comments.

---

## [Editor Report]

Thank you for revising the manuscript. There are a few remaining comments from the first reviewer.